# Anti-tumor effects of P-LPK-CPT, a peptide-camptothecin conjugate, in colorectal cancer

Lidan Hou[1,2,3,7], Yichao Hou[1,2,3,7], Yu Liang[1,2,3,7], Baiyu Chen[4], Xintian Zhang[1,2,3], Yu Wang[1,2,3], Kun Zhou[1,2,3], Ting Zhong[1,2,3], Bohan Long[1,2,3], Wenjing Pang[1,2,3], Lei Wang[1,2,3], Xu Han[1,2,3], Linjing Li[1,2,3], Ci Xu[1,2,3], Isabelle Gross[5,6], Christian Gaiddon[5,6], Wei Fu [4✉], Han Yao [1,2,3✉] & Xiangjun Meng [1,2,3✉]

To explore highly selective targeting molecules of colorectal cancer (CRC) is a challenge. We previously identified a twelve-amino acid peptide (LPKTVSSDMSLN, namely P-LPK) by phage display technique which may specifically binds to CRC cells. Here we show that P-LPK selectively bind to a panel of human CRC cell lines and CRC tissues. In vivo, Gallium-68 ([68]Ga) labeled P-LPK exhibits selective accumulation at tumor sites. Then, we designed a peptide-conjugated drug comprising P-LPK and camptothecin (CPT) (namely P-LPK-CPT), and found P-LPK-CPT significantly inhibits tumor growth with fewer side effects in vitro and in vivo. Furthermore, through co-immunoprecipitation and molecular docking experiment, the glutamine transporter solute carrier 1 family member 5 (SLC1A5) was identified as the possible target of P-LPK. The binding ability of P-LPK and SLC1A5 is verified by surface plasmon resonance and immunofluorescence. Taken together, P-LPK-CPT is highly effective for CRC and deserves further development as a promising anti-tumor therapeutic for CRC, especially SLC1A5-high expression type.

[1] Department of Gastroenterology, Shanghai Ninth People's Hospital, Shanghai Jiao Tong University School of Medicine, Shanghai, China. [2] Center for Digestive Diseases Research and Clinical Translation of Shanghai Jiao Tong University, Shanghai, China. [3] Shanghai Key Laboratory of Gut Microecology and Associated Major Diseases Research, Shanghai, China. [4] Department of Medicinal Chemistry, School of Pharmacy, Fudan University, Shanghai, China. [5] INSERM UMR_S1113, IRFAC, Strasbourg F-67200, France. [6] Universite de Strasbourg, Strasbourg 67200, France. [7] These authors contributed equally: Lidan Hou, Yichao Hou, Yu Liang. ✉email: wfu@fudan.edu.cn; hanyao89@163.com; meng_xiangjun@yahoo.com

Colorectal cancer (CRC) is the third most deadly cancer worldwide with over1.85 million cases and 850,000 deaths annually[1,2]. Presently, most patients are diagnosed at advanced stages with metastasis, but chemotherapy agents such as fluoropyrimidines, oxaliplatin, and irinotecan remain the major choice for patients with advanced CRC. Due to lack of tumor specificity, chemotherapies often result in poor efficacy and severe side effects. Hence, in order to overcoming these side effects, many active cytotoxic conjugates with anticancer activity are developed, however in most cases, their clinical application is still greatly limited by lack of selectivity[3]. The identification of specific tumor biomarkers for targeted therapies in recent years provide promising treatment options[4]. For example, the anti-VEGF (Bevacizumab) and anti-EGFR (Cetuximab) targeted therapies have been approved for the treatment of CRC and showed an encouraging achievement, but due to the frequent severe side effects, although their application in clinic is notably limited[5].

A promising alternative principle among various therapeutic strategies is to combine the tumor targeting vector with a cyto-toxic molecule to selectively kill tumor cells. Hence, the con-jugation of targeting molecules with chemotherapeutics, already used in clinics or in pre-clinical development, presumably enables selective delivery of cytotoxic payloads to target cells, resulting in an improved efficacy and a reduced systemic toxicity[6,7]. For example, researches based on targeting molecules such as proteins (monoclonal antibodies, peptides, etc.), nucleic acids (aptamers, etc.) have achieved encouraging results[8,9]. Recently, several antibody-drug conjugates (ADCs) have been approved as anti-cancer treatments and dozens more are in preclinical and clinical development[7].

The phage display technique is a powerful tool to identify novel targeting peptides in researches and clinics[10–12]. By screening whole cells, it is able to identify short peptides that interact with cell surface molecules. Using a M13 phage-display peptide library, we previously identified two peptides, CBP-DWS (DWSSWVYRDPQT) and P-LPK (LPKTVSSDMSLN), and both of them selectively bound to CRC cells[13]. Later studies found that although the CBP-DWS peptide showed specific binding ability to CRC cells, it did not improve the targeted killing ability of the chemotherapeutics (Camptothecin, CPT) when used as a targeted molecule to deliver the drug. Here, we report that the P-LPK peptide can be used as a targeting vector for drug delivery. When conjugated to CPT, the P-LPK-CPT conjugate exhibited improved anti-tumor effect with fewer side effects in vitro and in vivo, and the molecular mechanisms of P-LPK binds to CRC cell was also investigated.

## Results

### The P-LPK peptide selectively binding to CRC cell in vitro and ex vivo.
By screening a phage display library, a CRC specific binding peptide P-LPK (LPKTVSSDMSLN) was identified (Supplementary Fig. 1). To check if the P-LPK peptide is an actual target-binding peptide rather than a target-unrelated peptide (TUP), the sequence was blasted using the SAROTUP webserver[14], a suite of tools for examing possibly potential unrelated peptides (Supplementary Table 1). The results revealed that the P-LPK peptide did not match any previously identified TUP. Moreover, no complete homologous sequence was identi-fied in known protein banks, indicating that the peptide sequence was unique (Supplementary Table 2).

Next, the specific binding capacity of the P-LPK peptide to CRC cells was analyzed by confocal imaging of fluorescein isothiocyanate (FITC) labeled P-LPK conjugate (FITC-P-LPK) in vitro. A repertoire of human CRC cell lines, Colo320HSR,

HCT116, LoVo, HT29, SW480 were tested. In contrast to non-transformed human colonic epithelial cell line NCM460, the FITC-P-LPK conjugate showed abundant binding capacity to human CRC cell lines (Fig. 1a, Supplementary Fig. 2a), and the fluorescence intensity was significantly different between P-LPK and Control peptide P-CON (Fig. 1c, Supplementary Fig. 2b). The binding signal of the P-LPK peptide was also remarkably stronger in CRC tissues from six patients when compared to adjacent normal tissue (Fig. 1b, d). To further characterize the exact binding sites in cells, the P-LPK peptide was labeled with Rhodamine. As expected, the Rhodamine-P-LPK conjugate showed strong red fluorescence signal in HCT116 cells in contrast to NCM460 cells (Fig. 1e, Supplementary Fig. 2c). Moreover, the red signal of Rhodamine-P-LPK overlapped with the blue signal of the membrane probe, suggesting that P-LPK specifically binds in CRC cell membrane.

### MicroPET imaging of P-LPK binding to human CRC cells in vivo.
To validate the specific binding of the P-PLK peptide to CRC cells in vivo, the P-LPK peptide was labeled with a positron emitting radioisotope Gallium-68 ($^{68}$Ga) as a PET imaging tracer ($^{68}$Ga-P-CON as control). Two xenograft mouse models were included, the human malignant glioblastoma U87 cells and the CRC HCT116 cells, in order to obtaining coronal microPET images after injection of the $^{68}$Ga-P-LPK. A remarkable accu-mulation of the $^{68}$Ga-P-LPK in HCT116 tumors was observed in contrast to U87 tumors at 30 min, 60 min and 120 min after intravenous injection (Fig. 2a–c). A competitive binding assay was performed by pre-injection with the unlabeled free P-LPK peptide and the results revealed that the binding of $^{68}$Ga-P-LPK was dramatically attenuated to the HCT116 tumor (Fig. 2d) and the tumor/non tumor ratios (T/NT ratios) was significantly reduced, especially at 60 min and 120 min (Fig. 2e).

### The P-LPK-CPT displaying an enhanced selective anti-tumor activity in vitro.
Using P-LPK peptide as a targeting vector to delivery chemotherapeutic drugs toward CRC cells, we first checked whether the P-LPK peptide itself exhibits any biological activities. No cell proliferation differences was observed between the cells treated with the P-LPK peptide and the control peptide ($P > 0.5$) (Supplementary Fig. 3). The P-LPK peptide was con-jugated to the chemotherapeutic drug CPT by click chemistry to construct a conjugate which termed P-LPK-CPT (Fig. 3a–c), and the cellular uptake of the P-LPK-CPT conjugate was checked by confocal microscopy. The blue intrinsic fluorescence generated by the CPT was markedly observed in both cytoplasm and nuclei of LoVo, whereas was not in NCM460 cells, demonstrating that the P-LPK-CPT conjugate was selectively entering to the LoVo cells (Supplementary Fig. 4). These results indicated that the P-LPK-CPT harboring the potential to be further explored for improving therapeutic efficiency. The colony formation assay revealed that the P-LPK-CPT conjugate selectively and notably reduced the proliferation of HCT116 and LoVo cells, but not NCM460 cells (Fig. 3d). Cell proliferation activity was also measured by CCK8, which confirmed that the P-LPK-CPT conjugate greatly inhibited the proliferation activity of HCT116 and LoVo cells (Fig. 3e), and the EdU incorporation assay showed that the DNA synthesis was significantly suppressed in cells treated with the P-LPK-CPT conjugate (Fig. 3f). Altogether, the P-LKP-CPT conjugate was selectively taken by CRC cells, hence selectively exhibited an anti-proliferative activity.

### P-LPK-CPT conjugate selectively killing CRC cell in vivo.
The P-LPK-CPT conjugate was investigated for a targeted anticancer activity using mouse models. The P-LPK-CPT conjugate showed

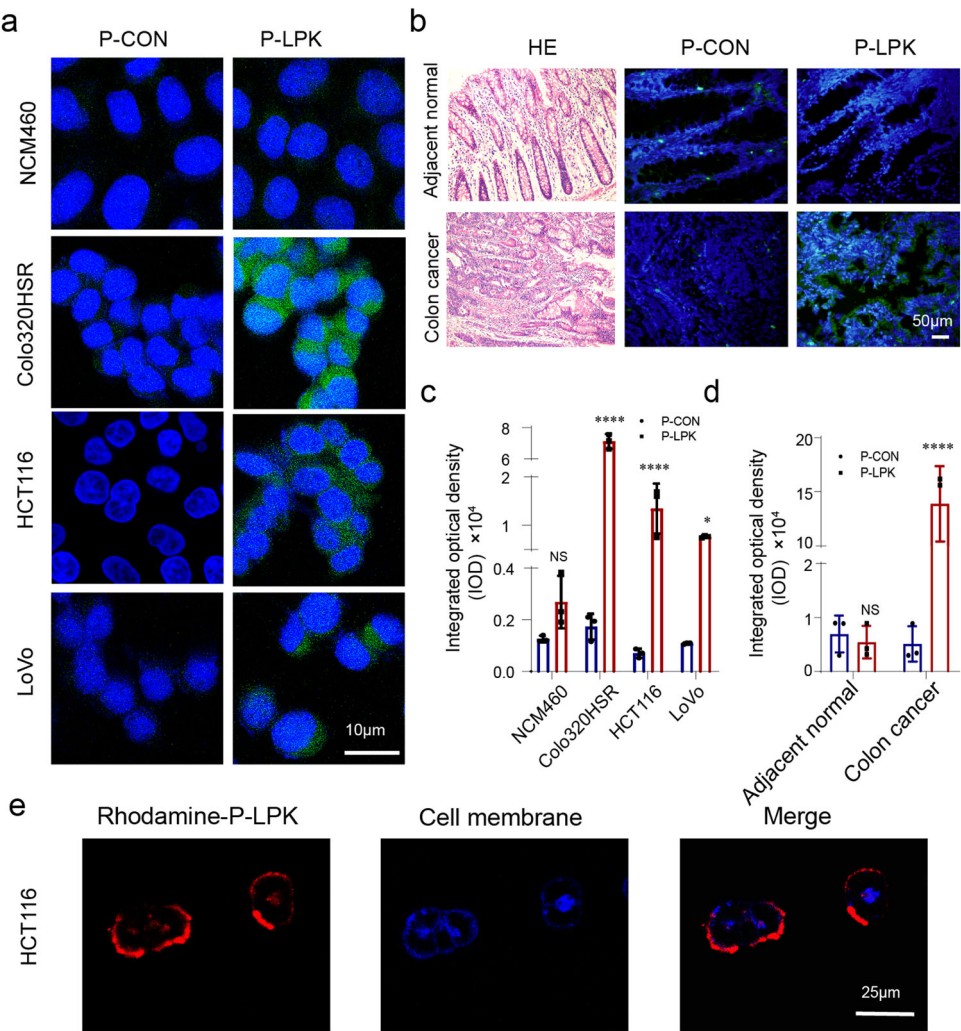

**Fig. 1 Fluorescent images of P-LPK binding to human CRC cells and cancer tissues. a** The FITC-P-LPK conjugate emitted stronger green fluorescence in CRC cells than normal NCM460 cells. Bar,10 μm. **b** The FITC-P-LPK conjugate selectively binds to CRC tissues (HE staining) Bar, 50 μm. **c, d** The fluorescence intensity of the P-LPK peptide in CRC cells (**c**) and tissues (**d**) was significantly higher than that of normal cells and tissues ($n = 3$, means ± SD, NCM460: P-CON vs P-LPK, $p = 0.9385$; Colo320HSR: P-CON vs P-LPK, $p < 0.0001$; HCT116: P-CON vs P-LPK, $p < 0.0001$; LoVo: P-CON vs P-LPK, $p = 0.0219$; Adjacent normal tissues: P-CON vs P-LPK, $p = 0.9934$;Colon cancer tissues: P-CON vs P-LPK, $p < 0.0001$) (*$p < 0.05$, ****$p < 0.0001$). **e** The binding site of the P-LPK peptide in HCT116 cells was investigated after labeling the peptide with Rhodamine. Bar, 25 μm.

a significant suppression of the tumor growth compared with control groups (***$P < 0.001$) (Supplementary Fig. 5). Besides, more than half of mice treated with CPT died of side effects during the experiment.

A patient-derived xenografts (PDXs) model was used to evaluate the therapeutic efficacy of the P-LPK-CPT conjugate. Consistent with the results of HCT116 xenograft model, the P-LPK-CPT conjugate markedly inhibited tumor growth compared to the P-CON-CPT conjugate, albeit no significant difference compared with CPT group (Fig. 4a, b). Interestingly, the tumor size of P-LPK-CPT-treated group did not show any significant decrease compared with CPT group, but histological analyses showed that CRC cells in the tumor tissues of the P-LPK-CPT-treated group was significantly reduced compared to CPT-treated group. Histologically, the cancer cells in tumor tissues of the P-LPK-CPT group were substantially consisted of connective tissues but CPT-treated group almost maintained its original morphological features (Fig. 4c). CPT accumulated in tumor tissues was evaluated and the blue intrinsic fluorescence signal of the CPT in the P-LPK-CPT-treated tumor tissues was significantly stronger than the control group, indicating that the P-LPK-CPT conjugate

was selectively accumulated in tumor tissues (Fig. 4d). These results supported the selective therapeutic efficacy of the P-LPK-CPT conjugate.

Gastrointestinal (GI) toxicity was evaluated in animals treated with the P-LPK-CPT conjugate was significantly lower compared to CPT alone (Fig. 4e, f). Organs such as heart, kidneys or lungs were investigated in all groups. The results showed that the conjugation of the CPT to the P-LPK peptide could reduce the damage to intestine and liver (Supplementary Fig. 6). To note also that routine blood parameters and biochemistry assays did not illustrate any significant toxicity in blood, liver and kidneys in P-LPK-CPT treated mice (Supplementary Fig. 7).

To further assess the anticancer activity of the P-LKP-CPT conjugate we measured relevant biomarkers for evaluating the pharmacodynamic effects of CPT[15,16]. Because CPT is a highly active topoisomerase I (Topo-I) inhibitor which causes rapid degradation of Topo-I through the ubiquitin-proteasome system and hence exhibits a strong anti-tumor activity[17−19], we therefore measured the Topo-I level in tumor tissues and found that Topo-I was significantly reduced in the P-LPK-CPT group compared to the P-CON-CPT group. In addition, the P-LPK-CPT conjugate

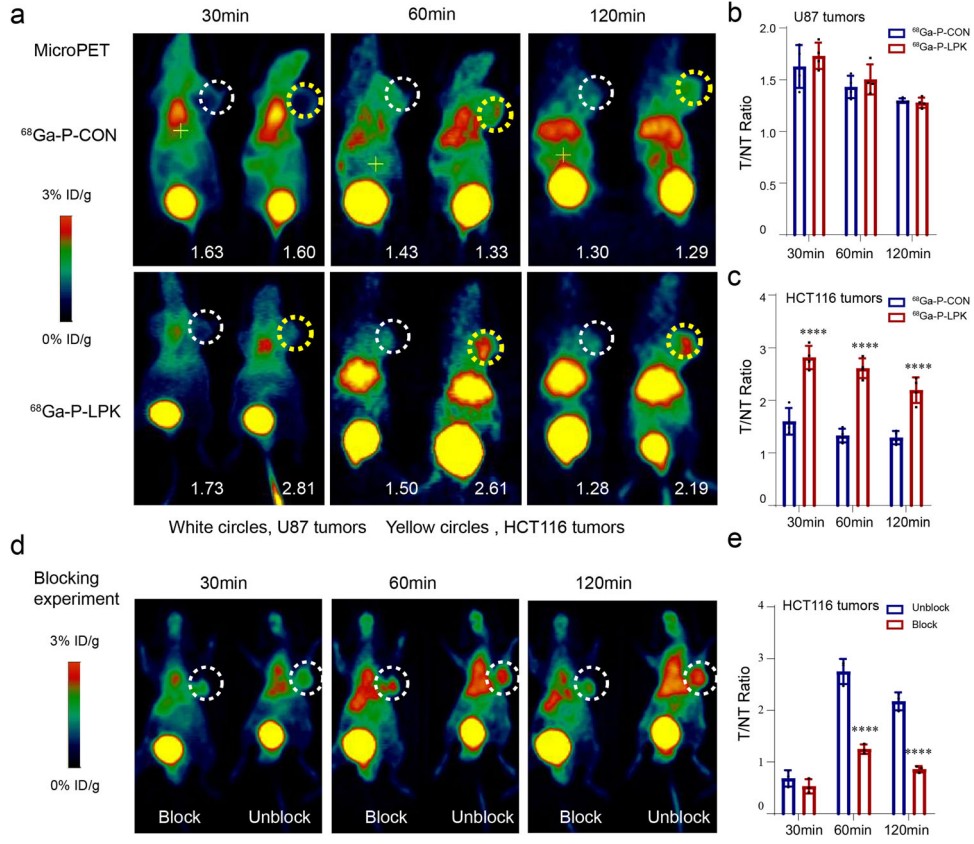

**Fig. 2 MicroPET imags of [68]Ga-P-LPK in U87 and HCT116 tumor–bearing nude mice. a** Whole-body coronal microPET images of U87 and HCT116 tumor-bearing mice ($n = 4$) at 30, 60, and 120 min after injection of [68]Ga-P-LPK or [68]Ga-P-CON (white circles, U87 tumors; yellow circles, HCT116 tumors). The quantification values of tumor were showed in the lower right corner of mice. **b**, **c** In contrast with [68]Ga-P-CON, the T/NT ratios of [68]Ga-P-LPK was significantly increased in the HCT116 tumor-bearing mice (**b**) but not in U87 tumor-bearing mice (**c**), suggesting that [68]Ga-P-LPK specifically binds to CRC tissues ($n = 4$, means ± SD, HCT116 30 min: [68]Ga-P-CON vs [68]Ga-P-LPK, $p < 0.0001$; 60 min: [68]Ga-P-CON vs [68]Ga-P-LPK, $p < 0.0001$; 120 min: [68]Ga-P-CON vs [68]Ga-P-LPK, $p < 0.0001$) (****$p < 0.0001$). The prominent uptake of [68]Ga-peptide was observed at the bladder at early time points (30 min), suggesting that this tracer may be excreted through the urinary route. **d**, **e** After blocking with unlabeled peptide, the T/NT ratios was significantly reduced at 60 min and 120 min ($n = 3$, means ± SD, 30 min: Unblock vs block, $p = 0.7512$; 60 min: Unblock vs block, $p < 0.0001$; 120 min: Unblock vs block, $p < 0.0001$) (****$p < 0.0001$).

markedly reduced the carbonic anhydrase IX (CAIX) staining, a marker for tumor hypoxia[15,20] (Supplementary Fig. 8a). P-LPK-CPT treatment also decreased proliferation and increased apoptosis of CRC cells (Supplementary Fig. 8b).

**SLC1A5 was identified as a major P-LPK receptor**. Beads conjugated with a streptavidin-biotin-P-LPK part were used to incubate with a protein extract from HCT116 lysates to perform a pull-down assay (Fig. 5a). A total of 737 proteins were found in the P-LPK condition, of which 241 proteins were relatively specific compared to the P-CON condition (Fig. 5b, Supplementary Data file 1). The most possible binding partner for the P-LPK peptide was analyzed from which 4 proteins (ATP2A2, SLC1A5, ANXA3, CACNA2D1) were interested (Supplementary Data file 2). Further analyses revealed that CACNA2D1 is in endoplasmic reticulum membrane and ATP2A2 in sarcoplasmic or endoplasmic reticula membrane. SLC1A5 and ANNX3 are in the cell surface that might represent possible receptors of P-LPK (Fig. 5b).

**Molecular interactions between P-LPK and candidate receptors**. The interaction between P-LPK peptide and SLC1A5 or ANXA3 was investigated via molecular docking method (Supplementary Fig. 9a). First, the BLAST program was used to search

a template protein for modelling the structure of the P-LPK peptide. The fairly homologous sequence of the P-LPK peptide was included in the complex structure of the tRNA methyl-transferase Trm1 from Aquifex aeolicus (PDB code: 3AXS, with a resolution of 2.16 Å). The P-LPK peptide showed about 67% sequence identity with the Trm1, which provided a solid template for modeling the 3D structure of the P-LPK peptide. The Discovery Studio 2019 program was employed to assemble the 3D models of the peptide using the structure of the Trm1 as a template. The FASTA program (http://www.ebi.ac.uk/Tools/fasta/index.html) was used to conduct sequence alignment. Residues in most favored regions in the Ramachandran plot were 77.8% and residues in additional allowed regions were 22.2%, suggesting that the 3D structure of the P-LPK peptide was accurate (Supplementary Fig. 9b).

Next, the X-Ray structures of SLC1A5 (pdb code: 6gct) and ANXA3 (pdb code:1AXN) were characterized and downloaded from PDB database. The disulfide bonds between the N-terminal and C-terminal in SLC1A5 and ANXA3 were created by Discovery Studio 2019 program. The docking program Hex v8.0[21] was used for the preliminary protein-protein docking, and the structure-prediction-based RosettaDock with Monte Carlo-based algorithm[22] was used to probe the possible minimum-energy conformations of the peptide binding sites of SLC1A5 and ANXA3. The estimated ΔG is −6.7770 kcal/mol and

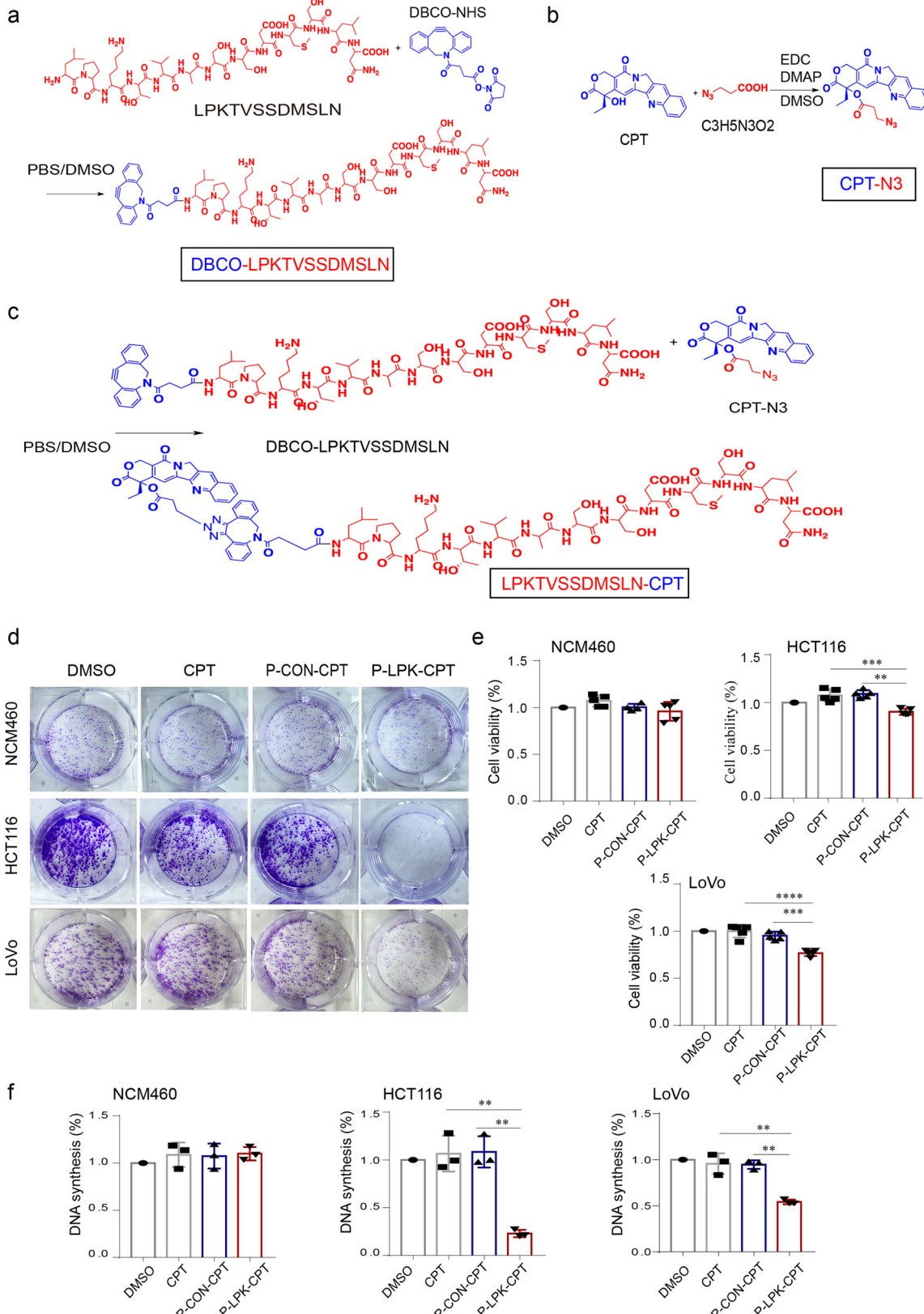

**Fig. 3 In vitro cytotoxicity of P-LPK-CPT in CRC cells. a–c** Illustration of how the P-LPK-CPT conjugate was synthesized (Details reference to materials and methods). **d** The colonal forming capability after different treatments with the P-LPK-CPT conjugate of HCT116, LoVo and NCM460 cells. **e** Cell proliferation was measured by CCK-8 ($n = 5$, means ± SD, HCT116: CPT vs P-LPK-CPT, $p = 0.0005$; P-CON-CPT vs P-LPK-CPT, $p = 0.0002$; LoVo: CPT vs P-LPK-CPT, $p < 0.0001$; P-CON-CPT vs P-LPK-CPT, $p = 0.0004$) (**$p < 0.01$, ***$p < 0.001$, ****$p < 0.0001$). **f** The proportion of cells in the DNA synthesis state was qualified and one-way analysis of variance is used to test the difference between groups ($n = 3$, means ± SD, HCT116: CPT vs P-LPK-CPT, $p = 0.0017$; P-CON-CPT vs P-LPK-CPT, $p = 0.0015$; LoVo: CPT vs P-LPK-CPT, $p = 0.0016$; P-CON-CPT vs P-LPK-CPT, $p = 0.0018$) (**$p < 0.01$).

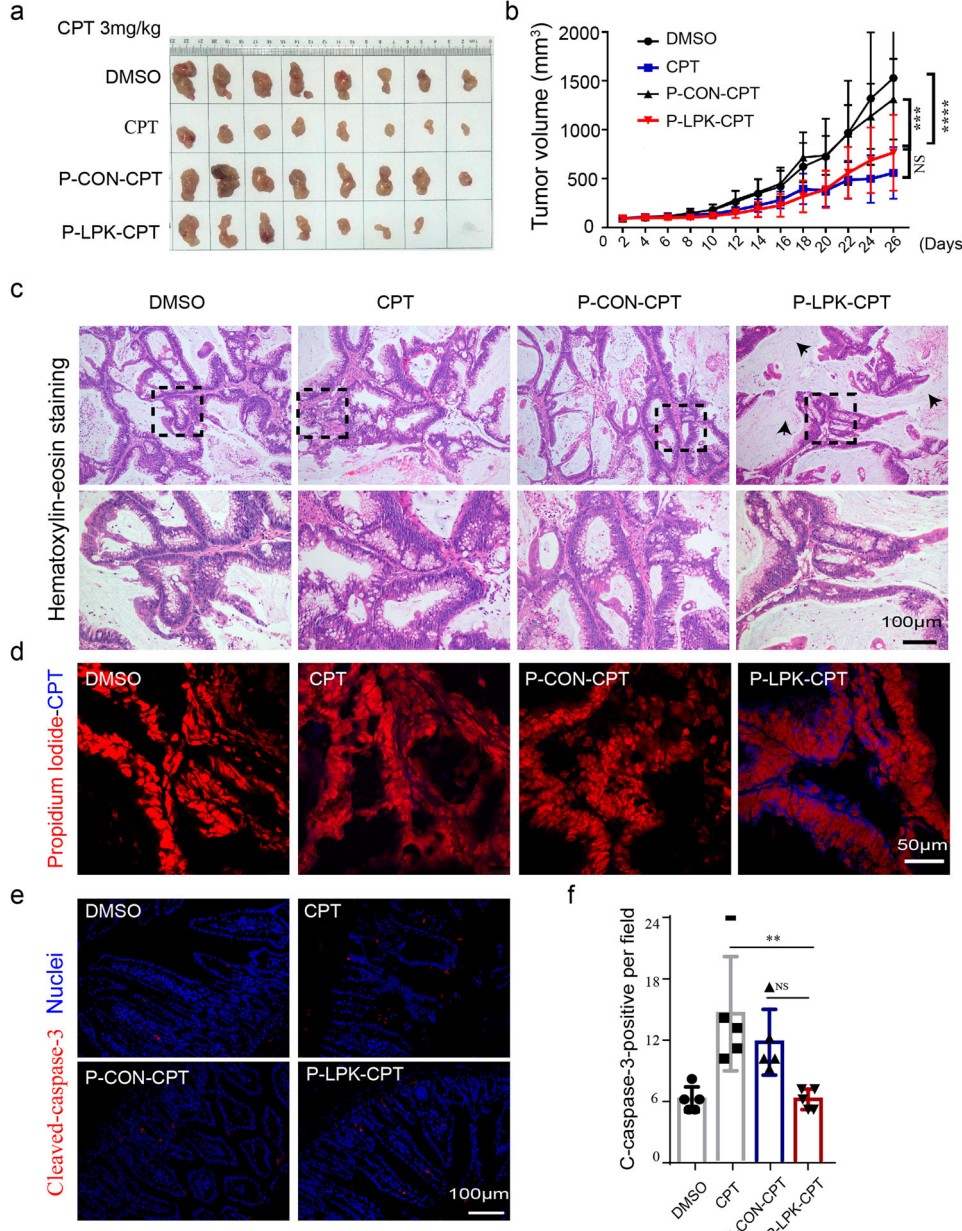

**Fig. 4 In vivo antitumor effects of P-LPK-CPT in CRC patient-derived xenografts. a, b** The size of tumors of different treatments after 26 days. Tumor volume changed after intravenous injection of the P-LPK-CPT conjugate ($n = 8$, means ± SD) (***$p < 0.001$, ****$p < 0.0001$). **c** HE staining showed that the tumor tissues of the P-LPK-CPT group were substantially replaced by connective tissues (arrowheads). **d** CPT signal (blue dots) was much stronger in the P-LPK-CPT tumor tissues. Red color was from propidium iodide (PI) for nuclear visualization. **e, f** Representative images (**e**) and quantification (**f**) of Cleaved caspase 3 immunofluorescence in colon tissues ($n = 5$, means ± SD) (CPT vs P-LPK-CPT, $p = 0.0051$; P-CON-CPT vs P-LPK-CPT, $p = 0.0722$; **$p < 0.01$).

−5.9317 kcal/mol, respectively. (Supplementary Fig. 9c). Solutions that violate available experimental binding data were discarded. Binding affinities were predicted by X-Score 1.3 with empirical scoring functions. The predicted affinities and all of available experimental data were used for the selection of the final complex.

The 3D structure models of the P-LPK peptide and SLC1A5 are shown in Fig. 5c. The hairpin-like structure of the peptide penetrated into a hydrophobic binding pocket of SLC1A5. The affinity and specificity of the P-LPK peptide was modulated by a series of local interactions. Two main kinds of interactions were observed: The side chain of the M267, S111, A423 and S354 formed hydrogen bonds with the L1, K3 and N12 in the P-LPK peptide. The residues L1, K3 and N12 in the P-LPK peptide inserted into the hydrophobic pocket formed by the L112, G115, S118, L119, I381, T384, V385, V426, G427, and A429 residues (Fig. 5d–f). ANXA3 displayed much lower affinity with P-LPK peptide than SLC1A5 (ΔG −5.9317 kcal/mol vs −6.7770 kcal/mol); probably due to only one hydrogen bond interaction was formed by the N12 in the P-LPK peptide and the K74, K98, K102, and Y137 residues (Supplementary Fig. 9d). Therefore, SLC1A5 could be the most possible receptor for P-LPK in CRC cells.

**Identification of SLC1A5 as receptor of P-LPK.** The direct binding of the P-LPK peptide to SLC1A5 was evaluated by Surface Plasmon Resonance (SPR). SLC1A5 protein was covalently

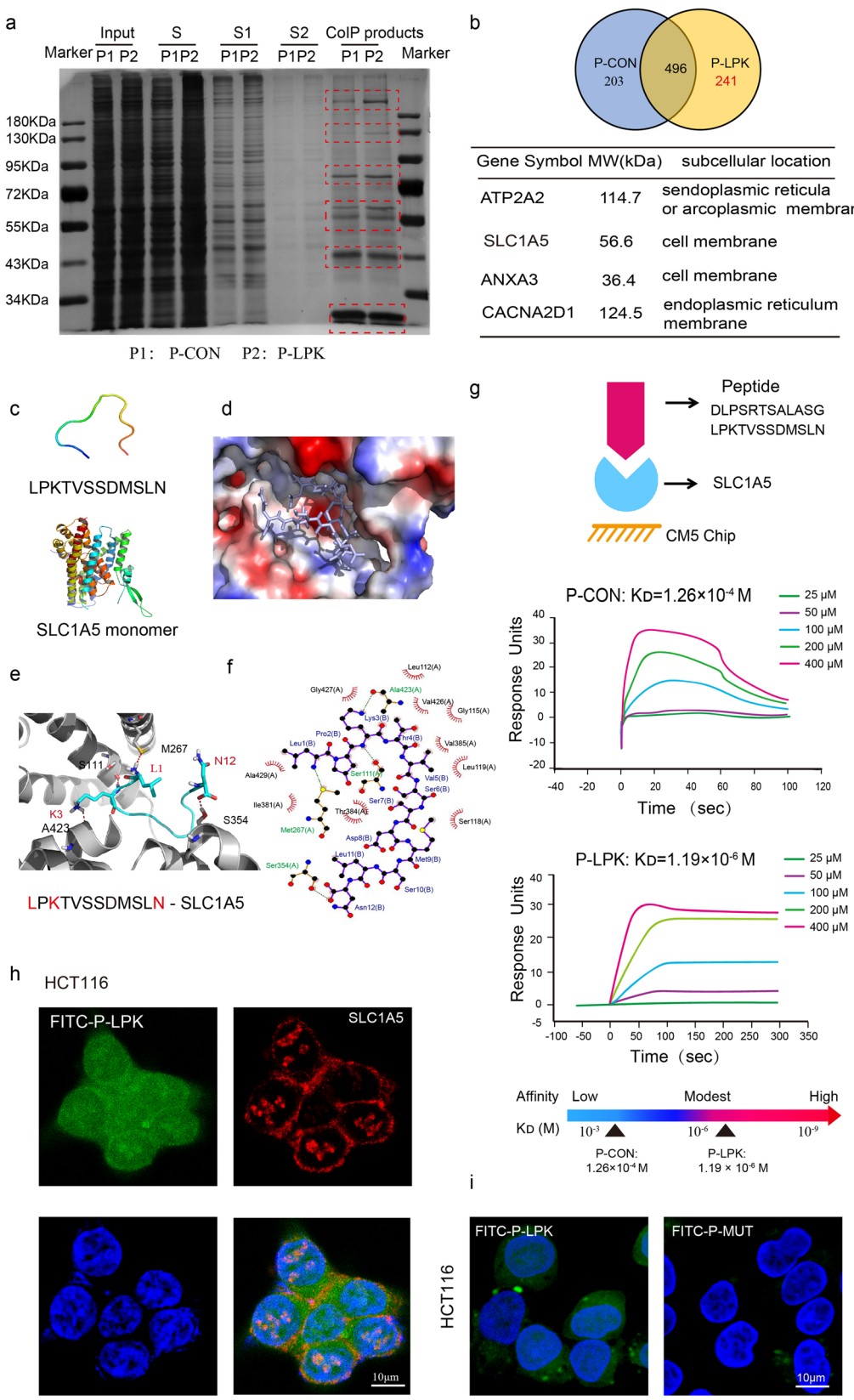

immobilized to the chip surface CM5. A representative sensorgram of peptide binding to the recombinant SLC1A5 was shown in Fig. 5g. The P-LPK peptide displayed a moderate binding affinity to SLC1A5, which was 100-fold compared to the control P-CON peptide ($K_D$ of $\sim 1.19 \times 10^{-6}$ vs. $\sim 1.26 \times 10^{-4}$ M).

The localization of SLC1A5 and the P-LPK peptide predominantly was in cell membrane and co-localized by dual immunostaining (Fig. 5h). Subsequently, P-LPK was mutated (Sequence APATVSSDMSLA) and the interaction with SLC1A5 was detected by immunofluenrence. The fluorescence intensity of FITC labeled mutant peptide (FITC-P-MUT) bound to HCT116

**Fig. 5 SLC1A5 was identified as the receptor of P-LPK. a** P-LPK-specific binding proteins were investigated via Co-immunoprecipitation test. 50 µg/ml Biotin-P-CON (P1) or Biotin-P-LPK (P2) was incubated with HCT116 at 37 °C for 2 h. Then cells were lysed and centrifuged at 12,000 rpm for 20 min at 4 °C. Differential protein bands between P1 and P2 were indicated by the red dashed box. (Details reference to materials and methods). S, supernatant group; S1, supernatant group 1; S2, supernatant group 2; co-IP products, the beads-P-LPK-specific binding protein compounds. **b** Targeted proteins were detected by mass analysis and several closely related P-LPK-specific binding proteins were focused. **c** The 3D structure models of the P-LPK peptide and SLC1A5. Molecular interactions of P-LPK to SLC1A5 were predicted by molecular docking. **d–f** The binding pocket of SLC1A5 was represented by (**d**) the electrostatic potential, where P-LPK were shown in pale blue stick (**e**) ribbon, where P-LPK was shown in cyan (**f**) 2D topological binding mode by Ligplot + (v2.2), in which hydrogen bonds were shown in green dotted lines and hydrophobic interaction were shown in red edges. **g** The affinity of P-LPK for SLC1A5 was measured by surface plasmon resonance (SPR) and expressed as resonance units (RU). **h** Co-localization of SLC1A5 (Red) and P-LPK (Green) was observed by dual immunostaining. Bar,10 µm. **i** P-LPK was mutated (Sequence APATVSSDMSLA) and the fluorescence intensity of FITC labeled mutant peptide (FITC-P-MUT) bound to HCT116 cells. Bar,10 µm.

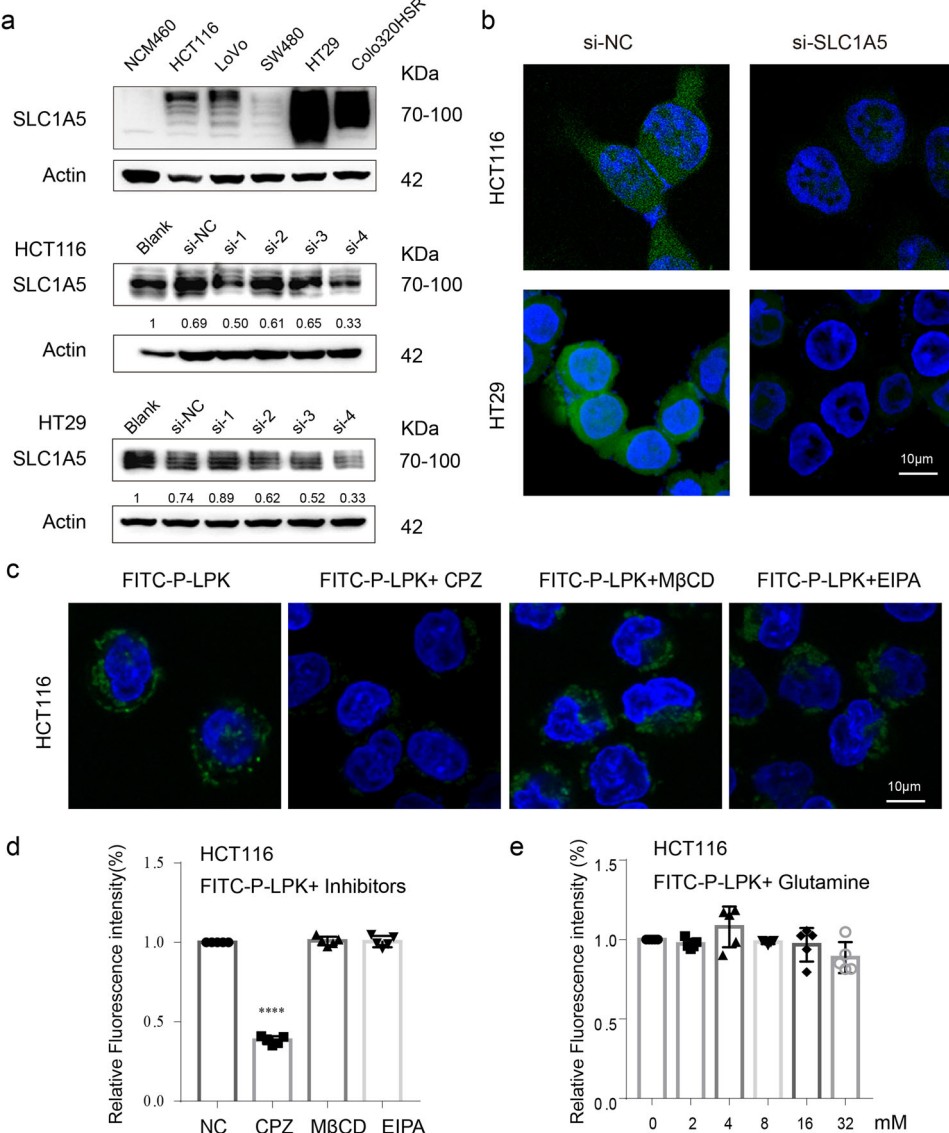

**Fig. 6 P-LPK bound to SLC1A5 and entered CRC cells through endocytosis. a**, **b** Western blotting analysis of SLC1A5 in NCM460, HCT116, LoVo, SW480, HT29 and Colo320HSR cells. After knocking down SLC1A5 in HCT116 and HT29, the binding intensity of FITC-P-LPK to cells was observed by confocal microscopy. Bar, 10 µm. **c**, **d** Effects of different endocytosis inhibitors on the internalization of FITC-P-LPK. HCT116 cells were pre-treated with 10 µM chlorpromazine, 50 µM methyl-β-cyclodextrin, 20 µM Amiloride hydrochloride for 30 min at 37 °C. Subsequently, the cells were incubated with FITC-P-LPK for 2 h. Then the fluorescence intensity was observed by confocal microscopy and detected using a multi-well plate reader ($n = 5$, means ± SD, NC vs CPZ, $p = 0.0001$; NC vs MβCD, $p = 0.9069$; NC vs EIPA, $p = 0.9812$) (****$p < 0.0001$). Bar, 10 µm. **e** The competition experiment with SLC1A5 substrates glutamine. FITC-P-LPK was incubated with increasing concentrations (0–32 mM) of glutamine in HCT116 cells at 37 °C for 2 h. The fluorescence intensity was measured in a multi-well plate reader ($n = 5$, means ± SD).

cells was significantly reduced (Fig. 5i). Furthermore, data from CRC TCGA and GEO database (Accession number is GSE41258) showed that SLC1A5 was remarkably unregulated in CRC compared with adjacent non-tumorous tissues (Supplementary Fig. 9e). The SLC1A5 expression level was significantly higher in a panel of CRC cell lines (HCT116, LoVo, SW480, HT29 and Colo320HSR), but not in normal NCM460 (Fig. 6a). After SLC1A5 was knockdown in HCT116 and HT29, the FITC-P-LPK showed a reduction of the fluorescence (Supplementary Fig. 10 and Fig. 6a, b), further more validated the role of SLC1A5 in the binding of P-LPK to CRC cells.

Macromolecules entry into cells is mainly through different endocytic mechanisms, such as clathrin-mediated endocytosis, macropinocytosis. Therefore, we hypothesized that the peptide P-LPK binds to SLC1A5 and then enters the cell through endocytosis. To test this hypothesis, we performed endocytosis inhibition experiments. HCT116 cells were pretreated with various endocytotic inhibitors, including chlorpromazine (CPZ, inhibiting clathrin-mediated endocytosis), methyl-betacyclodextrin (MβCD, inhibiting lipid raft-mediated endocytosis) and amiloride (EIPA, inhibiting macropinocytosis) and then incubated with FITC-P-LPK. The fluorescence intensity was observed by confocal microscopy and detected using a multi-well plate reader. The results showed that chlorpromazine remarkably decreased the cellular internalization of FITC-P-LPK, suggesting that the uptake of P-LPK was predominantly via a clathrin-mediated mechanism (Fig. 6c, d). To evaluate if P-LPK was transported by the classical elevator mechanism of SLC1A5, the competition experiment with SLC1A5 substrates glutamine was performed. FITC-P-LPK was incubated with increasing concentrations (0–32 mM) of glutamine in HCT116 cells. The intensity of FITC fluorescence did not change along with increasing glutamine concentration, indicating that glutamine does not affect the uptake of P-LPK (Fig. 6e).

## Discussion

The present study identified a peptide, P-LPK, which has the capacity of selectively binding to CRC cells, and this property was employed in a P-LPK-camptothecin conjugate. We found P-LPK remarkably improves the antitumor effects of CPT, and a glutamine transporter SLC1A5 was identified as the receptor of the P-LPK peptide.

The phage display technology is an effective method for screening peptides that may bind to a targeted cell. Multiple peptides targeting a wide panel of cancers have already been identified recently[23]. In the present study, the P-LPK peptide was identified from a phage display peptide library as an unreported peptide CRC specific binding peptide which is free from any risk of being a TUP (Supplementary Table 1 and 2).

However, the phage-displayed peptide specifically binds to CRC cells do not consequently mean that the peptide synthesized chemically still keep this property. Here, we found the synthesized peptide still maintains specific binding capacity to human CRC in vitro and ex vivo, and demonstrated that the P-LPK peptide binds a receptor located at the membrane of CRC cell (Fig.1). Interestingly, in vivo xenografts experiments demonstrated that the P-LPK peptide displayed an enhanced tumor uptake, selectively accumulated in CRC cells but not in glioblastoma cells, suggesting that P-LPK selectively targets CRC cells (Fig. 2).

Theoretically, exploiting peptides as drug-targeting tools to deliver chemotherapeutics to cancer cells can enhance drug efficacy and decrease side effects. However, it would be inappropriate to be used as a tumor-targeting vector if the peptide itself harbors any sort of biological activities, such as stimulating cancer cell growth. Encouragingly, the P-LPK peptide itself does not show any biological activities and therefor can be used as a valuable tool for targeted therapy (Supplementary Fig. 3).

A precondition of conjugating peptide to any drugs requires chemical groups on both of the peptide and the drug that are compatible for chemical conjugation and any binding will not hurt the drug's anti-cancer activity. Ideally, the peptide-drug bond remains stable under normal conditions, but the payload drugs should be easily released once the conjugated compound reaching its target. It is reported that several different chemotherapeutic drugs have been conjugated to cancer-specific peptides, including epirubicin (EPI), paclitaxel (PTX), and CPT[24–26].

Here, we successfully conjugated the P-LPK peptide to CPT by click chemistry and we termed the conjugate P-LPK-CPT (Fig. 3a–c). We again confirmed the selective uptake of the P-LPK-CPT conjugate by CRC cells, and it exhibited a selectively cytotoxicity to CRC cells (Supplementary Fig. 4). Hence, we speculate that after P-LPK-CPT is internalized into CRC cells, CPT can be released owing to the hydrolysis of the ester bond by the esterase that is highly expressed in CRC cells and favored by the acidic tumor microenvironment[27–29].

The anti-tumor potential of P-LPK-CPT was confirmed in two different models of xenografts. In both models, the P-LPK-CPT treatment showed a significant suppression of the tumor growth. It is worth noting that more than half of the mice died during CPT treatment, possibly due to the strong adverse effects of CPT[30]. In contrast, the P-LPK-CPT groups did not show significant change in the morphology or physiological markers of the different organs, highlighting the safety improvement brought by the targeting peptide.

The specific binding mechanisms of a peptide to a cell is presumably by its binding to a given receptor expressed at the cell membrane. For example, the receptor for the NGR peptides in tumor vasculature is aminopeptidase N[31]. Two peptides, PKRGFQD and SNTRVAP, which targeted the cell surface of androgen-independent prostate cancer cells, bind specifically to α-2-macroglobulin and 78 kDa glucose-regulated protein, respectively[32]. Bioinformatics analysis indicate that peptides PRWAVSP and DTFNSFGRVRIE specifically targeted Metalloproteinase Inhibitor 1 (TIMP-1) and Plasminogen activator inhibitor 1 precursor (PAI1) in MDA-MB-231, respectively[33].

Based on co-immunoprecipitation and molecular docking analysis, we identified SLC1A5 as the receptor of the P-LPK peptide. SLC1A5, a key glutamine transporter, is a sodium-dependent transporter that exchanges neutral amino acid across cell membrane of peripheral tissues. It was found that SLC1A5 was highly expressed in CRC[34,35]. We also detected SLC1A5 in U87 cells and found the expression level of SLC1A5 in U87 cells was significantly lower than HCT116 cells, which is consistent with the microPET imaging results. Moreover, the binding of P-LPK and SLC1A5 was confirmed by surface plasmon resonance and immunofluorescence. In addition, the solute carrier family 1 (SLC1) consists of five high-affinity glutamate transporters EAAT1-EAAT5 and two neutral amino acid transporters ASCT1 and ASCT2 (ASCT2, encoded by gene *SLC1A5*). Although SLC1 family members have similar predicted structures, they exhibit distinct functional properties. In addition to its role as a transporter, SLC1A5 is also a receptor for many retroviruses, including simian retrovirus 4, feline endogenous virus, etal. Here, we found that SLC1A5 might be the receptor of P-LPK and helps the entry of the peptide P-LPK. However, P-LPK did not cross membrane as a "substrate" of SLC1A5 considering that glutamine did not compete with the peptide P-LPK for the binding to SLC1A5 (Fig. 6). Theoretically, P-LPK may have cross binding ability to other SLC1 families, but further experimental verification is required.

Altogether, our results demonstrate that the P-LPK peptide was a potential weapon for targeting CRC. Still, several aspects remain to be investigated, for instance, the immunogenicity of P-LPK-CPT conjugate was not yet evaluated, which should be considered before a biomedical application, and more details of the binding site to SLC1A5 needs to be elucidated via mutation experiment, et. al.

## Methods

**Cell lines and cell culture.** The human CRC cell lines Colo320HSR, HCT116, LoVo, HT29, and SW480 were all obtained from the American Type Culture Collection (ATCC, Manassas, VA). Normal human intestinal epithelial cells NCM460 was purchased from the Chinese Academy of Sciences, Shanghai Branch. Cells were cultured in RPMI-1640, F12K, DMEM, Mycoy's 5a (Invitrogen) with 100 U/mL penicillin, 100 g/mL streptomycin and 10% fetal bovine serum (Grand Island, NY, USA).

**Animal models.** Female BALB/c nude mice (4–6 weeks with weights of 18–22 g) were obtained from Shanghai Super-B&K Laboratory Animal Corp. Ltd. (Shanghai, China) and kept in specific pathogen free, temperature-controlled isolation. All animal studies were conducted in accordance with the principles and procedures outlined in the Guide for the Care and Use of Laboratory Animals and were approved by Ethics committee. To generate the HCT116 and U87 tumor model, about $1 \times 10^6$ tumor cells were subcutaneously injected into the right front flank of 4 weeks old athymic nude mice.

PDX models were established. Briefly, the patient-derived CRC xenograft model was established from a biospecimen obtained from a patient undergoing primary disease resection. CRC samples were pieced into 25–30 mm$^3$ in PBS with 100 mg/ml penicillin and 100 U/ml streptomycin. Tumor pieces with Matrigel (BD Biosciences) were transplanted into the right flanks of 6-week-old mice. After tumor formation, tumors were expanded through 3–4 passages until tumors were sufficient for implantation.

**Imaging of fluorescent peptide binding to cancer cells.** NCM460, Colo320HSR, HCT116, LoVo, HT29 and SW480 cells were respectively incubated with FITC-labeled P-LPK (Random peptide P-CON) peptides for 2 h at 37 °C and washed with PBS for three times. The cells were fixed with 4% paraformaldehyde for 30 min at room temperature, and then 4,6-diamidino-2-phenylindole (DAPI) was used for nucleic acid stain. A laser scanning confocal microscope was used to visualize the cells. The fluorescence intensity was analyzed using the Image-Pro Plus 6.0 software. For cell localization experiment, 20 µg/ml rhodamine-labeled peptides were incubated with NCM460 and HCT116 for 2 h, and then washed with PBS for three times. After that, the cell membrane was stained with a blue cell membrane probe (Thermo Fisher Scientific, Cat. No. W11263). The binding of Rhodamine-P-LPK to cells and their localization in cells was observed under a confocal microscope.

**Radiosynthesis of $^{68}$Ga-peptide.** To obtain $^{68}$Ga, 0.1 M HCl was used to elute a $^{68}$Ge-$^{68}$Ga generator. The radioactivity of eluate was monitored. The radioactive elution peak was collected for $^{68}$Ga radiolabeling experiment. An aliquot of 150 µg of compound P-LPK or P-CON was put in a glass vial and dissolved with 0.2 M sodium acetate/acetic acid buffer, pH 3.8. Then $^{68}$GaCl$_3$ tracer was added, and the pH was adjusted to fall in a range of 3.5~4 with 0.1 M NaOH. The solution was stored at room temperature for 0.5 h. The radiochemical purities of $^{68}$Ga-P-LPK and $^{68}$Ga-P-CON were measured with thin-layer chromatography (TLC), using silica gel-impregnated glass fiber stripe (iTLC SG) and 0.25 M sodium citrate/citric acid buffer as mobile phase. $^{68}$GaCl$_3$ was used as a control.

**MicroPET imaging.** Each U87 and HCT116 tumor-bearing mouse was injected in a tail vein with 3.7 MBq (100 µCi, 0.1 nmol in 100 µl) of $^{68}$Ga-P-LPK or $^{68}$Ga-P-CON under isoflurane anesthesia ($n = 4$ per group). For static PET, 5-min scans were acquired at 30 min, 60 min, and 120 min after injection. The maximum radioactivity concentrations within a tumor or an organ were obtained from mean pixel values within the multiple regions of interest (ROI) volume.

For the blocking experiment, unlabeled P-LPK was first injected into mice ($n = 3$ per group), and 3.7 MBq $^{68}$Ga-P-LPK was then injected. Static PET images were acquired at 30 min, 60 min and 120 min post-injection.

**Synthesis of the P-LPK-CPT conjugate.** The DBCO-NHS ester was dissolved in DMSO, and the P-LPK peptide solution was added drop wise. The mixture was stirred overnight to prepare a liquid phase to separate and purify the DBCO-peptide. CPT and azidopropionic acid C$_3$H$_5$N$_3$O$_2$ were suspended in anhydrous dichloromethane for DCC condensation, and then complete the esterification reaction. Purified by silica gel column and eluted with dichloromethane/methanol to obtain azide-modified camptothecin (CPT-N$_3$) The CPT-N$_3$ was dissolved in DMSO and purified peptide- CPT was prepared by reaction with DBCO- peptide.

**In vitro cytotoxicity of P-LPK and anti-tumor effects of P-LPK-CPT.** The in vitro cytotoxic effects of the P-LPK peptide were evaluated by a standard Cell Counting Kit-8 (CCK8) colorimetric assay. Normal human intestinal epithelial cells (NCM460) and human CRC cells (HCT116 and LoVo) were separately seeded into 96-well cell culture plates and incubated overnight. The P-LPK peptide at different concentrations was added to the wells. The cells were subsequently incubated in the same condition for 24 h, 48 h, and 72 h. Finally, the culture medium solutions of CCK8 were added to each well, and the cells were incubated for 2 h. For the in vitro anticancer effects of the P-LPK-CPT conjugate, the NCM460, HCT116 cells and LoVo cells were cultured in media containing DMSO, free CPT, P-CON-CPT, and P-LPK-CPT with equivalent CPT dose of 10 nM for 48 h. The cell viability was detected by the absorbance at 450 nm using a microplate reader.

**The 5-ethynyl-2′-deoxyuridine incorporation assay.** 96-well plates were seeded at 3000 cells per well and exposed to different treatments. 50 µM EdU labeling medium (RiboBio) was added and incubated for 2 h at 37 °C. The cells were then fixed with 4% cold methanol for 10 min. After being washed with PBS, cells were stained with anti-EdU working solution for 30 min and stained with Hoechst33342 for 10 min. Cells were observed using fluorescent microscopy. The percentage of EdU-positive cells was calculated from five random fields in three wells.

**Colony formation assay.** NCM460, HCT116 and LoVo cells were seeded in 6-well plates at a density of ~200 cells per well. Cell culture medium containing DMSO, free CPT, P-CON-CPT, and P-LPK-CPT (with the equivalent CPT doses of 10 nM) was added for 24 h. Then the cells were washed with PBS for three times and incubated for another 10 days. Subsequently, cells were fixed with 4% paraformaldehyde and stained with crystal violet solution. Above 50 cells were counted as one colony under a microscope.

**Therapeutical evaluation of P-LPK-CPT in vivo.** The HCT116 tumor-bearing mice were randomly divided into 4 groups ($n = 7$ for each group). After the tumor size reached about 100 mm$^3$, the mice were intravenously injected twice a week with DMSO, free CPT, P-CON-CPT, and P-LPK-CPT (Equivalent CPT dose of 5 mg/kg). The tumors were measured by a vernier caliper twice a week. The weights of the mice were recorded. Then, the mice were sacrificed at the 28th day. The tumors were dissected and weighted. The tumor volumes were calculated to evaluate the therapeutic efficacy.

PDX mice were also randomly divided into four groups ($n = 8$) and injected with DMSO, free CPT, P-CON-CPT, and P-LPK-CPT (Equivalent CPT dose of 3 mg/kg) twice a week. The tumors were measured every other day and harvested at 26th day. The tumor tissues were immune-stained for Topo-I (Proteintech, Wuhan, China, Cat No.20705-1-AP, 1:100) and CAIX (Proteintech, Wuhan, China, Cat No. 11071-1-AP, 1:100), Ki67 (Servicebio, Wuhan, China, Cat No. GB111499, 1:500) and cleaved caspase-3 (Servicebio, Wuhan, China, Cat No. GB11532, 1:500). Mice intestinal tissues were collected, processed, and stained for cleaved caspase-3.

**Tissue confocal immunofluorescence microscopy.** Mice tumor tissues used for fluorescence were immediately frozen in liquid nitrogen and cut into 7 µm-thick frozen sections. The slide was mounted with DAPI. A confocal microscope was used to collect the images (CPT-excitation: 370 nm, emission: 440 nm).

**Co-immunoprecipitation.** 50 µg/ml Biotin-P-LPK (Biotin-P-CON) was incubated with HCT116 at 37 °C for 2 h. Then cells were lysed and centrifuged at 12,000 rpm for 20 min at 4 °C. The above supernatant was then suspended with pretreated magnetic beads overnight at room temperature, and the supernatant was collected for control (as supernatant group). The magnetic beads were eluted once with 200 µl RIPA lysis solution (First elution, as supernatant group 1) and captured with a magnetic stand (Invitrogen, USA), and the magnetic beads were eluted for the second time (Second elution, as supernatant group 2). Subsequently, the magnetic beads were added to 20 µl loading buffer and boiled at 100 °C for 10 min. Finally, the beads-P-LPK-specific binding protein compounds were collected and diluted with ddH$_2$O (Co-IP products). All collected protein complexes were eluted with 50 ml of ddH$_2$O by boiling for 10 min and then subjected to silver staining.

**Silver staining.** Equal amounts of proteins after co-immunoprecipitation were loaded in 8% SDS-PAGE and electrophoresed at 25 mA for 50 min. Gels were stained with Silver (P0017S, Beyotime, China) according to the manufacturer's instructions.

**Mass spectrum analysis.** The protein bands of interest were excised from Silver-stained gel. Each gel slice was diced into small pieces and placed into a 1.5 ml tube. A gel piece removed from a protein-free region of the gel was used as parallel control. Sample preparation used for EASY-nLC 1200 mass spectrometry was performed according to the standard protocol. Gel slices were de-stained and digested by trypsin overnight at 37 °C. The protein digests were later desalted for MS and MS/MS analysis, which were performed in our lab using the QExactive

system (Thermo Fisher Scientific, USA). Afterward, Proteome Discoverer software (version 1.4; Thermo Fisher Scientific, USA) was applied for protein identification and quantitation.

**Homology modeling**. P-LPK was built from tRNA methyltransferase Trm1 from Aquifex aeolicus (PDB code: 3AXS, with a resolution of 2.16 Å) as template using Discovery Studio 2019. Align Sequences module was utilized to align two sequences. All parameters were set as default. P-LPK was created using the Build Homology Models module in Discovery Studio 2019. Optimization level was set as high. Total Energy and lowest DOPE score was subjected to the further energy minimization.

**Docking and plotting**. Hex v8.0 was used for the preliminary protein-protein docking, and RosettaDock was further used to probe the possible minimum-energy conformations. The protein structures were visually checked and prepared by using the Prepare Protein Wizard encoded in Discovery Studio 2019. Protonation states of ionizable residues and histidine residues were predicted according to the microenvironment and pKa values calculated with the PROPKA algorithm (http://propka.org) at pH= 7.0. Missing side chains and hydrogen atoms for each structure were complemented and minimized with restraints. P-LPK and proteins were subjected to Hex v8.0 using the Shape + Electro + DARS Correlation Type. Other Parameters were set as default. Then the preliminary docking results were subjected to Flexpepdock of structure-prediction-based RosettaDock with Monte Carlo-based algorithm. All the Parameters were set as default. Pymol was employed in plotting.

**Protein-peptide binding assay using Biacore**. A surface plasmon resonance-based method using the Biacore T200 platform was developed to detect the binding of peptide (25–400 μM) to recombinant SLC1A5. The sensor chip CM5, 1 × PBS (2 mM KH2PO4, 10 mM Na2HPO4, 137 mM NaCl, 2.7 mM KCl) with 5% DMSO, 0.05% Tween 20 buffer (pH 7.4), and the amine coupling kit (which includes EDC, NHS, and ethanolamine reagents) were all obtained from GE Healthcare. The sensor chip CM5 was activated using EDC and NHS reagents, according to the standard procedure for amine coupling. 20 μg/ml SLC1A5 (about 10 μg proteins in 0.5 ml 10 mM Sodium Acetate buffer) was then covalently immobilized to the chip surface CM5 under 10 mM Sodium Acetate (pH 4.5) buffer. The resulting differences in resonance units (RU) were about 4000 RU. After de-activation by ethanolamine and washing with 1×PBS with 5% DMSO, 0.05% Tween 20 buffer (pH 7.4), the prepared sensor chip was applied to Biacore T200 for the binding assay. 1×PBS with 5% DMSO, 0.05% Tween 20 buffer (pH 7.4) was used as running buffer. A one-minute injection of a solution of 10 mM Glycine-HCl, pH 3.0 was used to regenerate the chip surface.

**Cellular localization assay**. HCT116 cells were fixed with methanol for 2 h and blocked with 5% BSA for 60 min at room temperature. After blocking, cells were incubated with FITC-P-LPK or anti-SLC1A5 (Abcam, Cat No. ab237704, 1:50) at 4 °C overnight. Then, anti-SLC1A5 cells were further incubated with APC conjugated anti-rabbit IgG secondary antibody (Thermo Fisher Scientific, Cat No. A-10931, 1:250) for 20 min at room temperature in a lucifugal chamber. Samples were washed with PBS for three times followed by treatment with DAPI. Samples were photographed using Zeiss LSM 710 laser confocal scanning microscopy.

**Endocytosis inhibition assay**. Chlorpromazine (CPZ), amiloride (AMI), and methyl-β-cyclodextrin (MCD) were purchased from MedChem Express. The cells were treated with different inhibitors (10 μM chlorpromazine, 50 μM methyl-β-cyclodextrin, 20 μM amiloride) for 30 min at 37 °C. Subsequently, the cells were incubated with FITC-P-LPK for 2 h. The uptake of FITC-peptide was stopped by washing three times in ice-cold PBS. Then the fluorescence intensity was detected using a multi-well plate reader (BioTek synergy4) and observed by confocal laser scanning microscopy.

**Glutamine competition experiment**. FITC-P-LPK was incubated with increasing concentrations (0–32 mM) of glutamine in HCT116 cells at 37 °C for 2 h. Then these cells were washed with ice-cold PBS three times gently, suspended in PBS and transferred to a 96-well costar plate. The fluorescence intensity was measured in a multi-well plate reader (BioTek synergy4).

**Statistics and reproducibility**. GraphPad Prism 7.0 was used for statistical analysis. All values are expressed as means±s.d. Student's t-test was used to compare between two groups. One-way analysis of variance (one-way ANOVA) with Tukey's post hoc tests was used for analysis between multiple groups. All experiments were performed in three replicates. In all tests, asterisks (*) denote statistically significant differences ($*p < 0.05$, $**p < 0.01$, $***p < 0.0001$, $****p < 0.0001$).
    Additional methods are provided in the Supplementary Material.

**Reporting summary**. Further information on research design is available in the Nature Portfolio Reporting Summary linked to this article.

## Data availability

All data associated with this study are included in the paper. The source data behind the graphs is presented in the Supplementary Data 3. Uncropped and unedited gel images are listed in Supplementary Fig. 12. The mass spectrometry proteomics data have been deposited to the ProteomeXchange Consortium with accession codes PXD03747. Any additional details are available from the corresponding author upon reasonable request.

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

## Acknowledgements

We thank Leilei Shi for helpful scientific discussions and technical advice. This work was supported by National Natural Science Foundation of China (Grant No.82003167, 81872419 and 32071377). Shanghai Sailing Program (19YF1426900, 20YF1424400). Interdisciplinary Program of Shanghai Jiao Tong University (ZH2018QNA01), ANR ITI InnoVec (CG, IG). Project of Biobank from Shanghai Ninth People's Hospital, Shanghai Jiao Tong University School of Medicine (No. YBKB201918).

## Author contributions

X.M. and H.Y. designed the study. X.M. supervised the project. X.M., I.G., and C.G. revised the manuscript. L.H., Y.H. and Y.L. performed the experiments and drafted the manuscript. W.F. and B.C. designed and performed the molecular docking analysis. H.Y., W.P. and L.W. provided technical assistance. X.Z., Y.W., K.Z., T.Z. and B.L. provided the human tissue samples. X.H., L.L. and C.X. participated in statistical analysis and data interpretation. All authors contributed to manuscript authoring and review.

## Ethics statement

The study was approved by Shanghai Ninth People's Hospital Affiliated Shanghai Jiao Tong University, School of Medicine Ethics committee. Animal subjects (Number SH9H-2020-A288-1).

## Competing interests

The authors declare no competing interests.
