## [Peer Review File · Communications Biology]

Reviewers' comments:

Reviewer #1 (Remarks to the Author):

the manuscript deals with an interesting topic that is the role of asct2 in CRC and with the potential therapeutic effects of a novel peptide. this is, of course, very interesting for both basic and applied research. however, some concerns arose which are listed below:

1) in general some sentences are not very clearly written such as pg 2 lines 50-51.

2) the hypothesis on the role of asct2 is not clear. what do the authors suggest? is asct2 responsible for binding the peptide or for transporting it? is the transport mediated by the classical elevator mechanism of asct2 or do the authors hypothesize an endocytotic mechanism based on ASCT2? consider that ASCT2 is considered a viral receptor not only a transporter, therefore the endocytotic mechanism needs to be taken into consideration. Indeed, the transport of a long peptide instead of a single amino acid is a bit difficult to think considering the mechanism of membrane transporters

3) the expression levels of ASCT2 in the tested cells lines should be demonstrated and shown by western blot analysis

4) a proof of peptide specificity, followed by docking, would be the use of a mutant peptide in those residues considered responsible for interaction with ASCT2

5) docking analysis should be redone because the authors used the 5lm4 structure that is not asct2 but eaat1. the cryoem structures of asct2 are available in different conformations, consider as an example pdb 6gct or others from the same authors.

6) another important experiment that could be done is the competition experiment with asct2 substrates to help distinguish between actual transport and another mechanism such as endocytosis

Reviewer #2 (Remarks to the Author):

This manuscript studies a peptide, P-LPK, which was identified from the phage-display library toward colorectal cancer cells (CRC). This study validates that P-LPK peptide binds to CRC cells in cell culture as well as in mice models. The authors further conjugate the peptide with camptothecin (CPT), a topoisomerase inhibitor as a chemotherapy agent. The P-LPK-CPT conjugate has cytotoxicity toward CRC cancer cells both in vitro and in vivo. The target of P-LPK was also investigated, and SLC1A5 is a potential target based on Co-IP and protein structure modeling. Overall, this is an interesting manuscript with the potential for future clinical application. However, the novelty is slightly compromised by the previous report showing another peptide, CBP-DWS, toward CRC from the same research group (Hou et al, 2018). Some of the experimental details or explanations are missing. The suggestions and questions are listed below.

1. The result that CPT alone is toxic to mice while P-LPK-CPT has an anti-tumor effect is interesting. The authors speculate that CPT is hydrolyzed from P-LPK-CPT by esterase in CRC. However, esterase activity is abundant in serum, for example, albumins have been shown to exhibit esterase activity. Can the authors provide explanations for why the P-LPK-CPT will not be hydrolyzed in blood and caused toxicity from CPT before the peptide reaches the tumor microenvironment?

2. Please soften the claim on title "targeting SLC1A5" and the last sentence of abstract "via targeting SLC1A5". Since the current data only support that P-LPK may bind to SLC1A5. If the author would like

- to keep the claim, please provide data that genetically or chemically inhibiting SLC1A5 did abolish the anti-tumor effect of P-LPK-CPT, while overexpression of SLC1A5 sensitizes the CRC cells to P-LPK-CPT.
3. Why the authors did not choose to conjugate P-LPK with the major CRC chemotherapy choices, such as 5-FU, oxaliplatin, and irinotecan?
 4. What does CBP- or P- stand for in the name of the two peptides, CBP-DWS and P-LPK?
 5. It is a puzzle why CBP-DWS-CPT doesn't have a cytotoxicity effect (Figure S2). Both CBP-DWS and P-LPK bind to CRC cells, but only P-LPK-CPT has a cytotoxicity effect. Can the author discuss the reasons that cause the discrepancy?
 6. Are Figure 1A and Figure S3A live images or after fixation? From the method section, it seems they are live images while the DAPI is listed. As we know, DAPI is used for fixed samples. What is the concentration of DAPI used?
 7. The fluorescence signals of SW480 P-CON vs P-LPK in Figure S3A are not very different, while the quantification data (Figure S3B) shows a significant difference. Please show representative images or please indicate how the quantification was processed. Also, please indicate if the microscopy setting and image process are identical between experimental and control groups.
 8. Please provide the catalog number of the Thermo "blue membrane probe".
 9. I am confused that the authors first claimed the peptide sequence of P-LPK is unique (supplementary table 2), but later they found P-LPK has a fairly homologous sequence with Trm1 (75%).
 10. Please show the data that P-LPK insets into the hydrophobic pocket (line 216, not shown).
 11. Please show the data to support the sentence " however, ANXA3 displayed much lower affinity...and the K74, K98, K102, and Y137 residues". (line 218-220).
 12. Which GEO database (accession number) is used for figure S10F?
 13. Please provide reference or data to support the statement "SLC1A5 is down-regulated in malignant glioblastoma".
 14. Typo errors: (1) line 354: DBCO- instead of DCBO- (2) line 366 and 367: ddH2O instead of ddH2O2.
 15. Figure 2 legend mentions that kidneys uptake 68Ga-peptide. Out of curiosity, why do the kidneys uptake a prominent amount of 68Ga-peptide? Also, it seems that the 68Ga-peptide accumulates at multiple tissues, not just the kidney (heart? Liver? Bladder?).
 16. Considering that CPT is a chemo drug and it induces toxicity in mice, wouldn't it be surprising that CPT alone and control-peptide-CPT don't have cytotoxicity effects (Figure 3D-3E)? Could the authors discuss more this finding?
 17. It is not fair to state that PI is a nuclear stain (Figure S5A figure legend) because PI also stains mitochondria DNA.
 18. The figure S10G is quite important and it may be moved to the main figure. Since Colo320HSR has the highest signal of P-LPK in figure 1A, does the cell line also have a high expression of SLC1A5?

Reviewer #3 (Remarks to the Author):

In this manuscript, the authors report the identification of a peptide-drug conjugate that specifically targets amino acid transporter SLC1A5 in colorectal cancer cells. Using a phage-display system the authors identify and validate the in vitro and in vivo safety profile and applicability of the novel peptide. They go on to show that the peptide specifically binds to cancer cells and accumulated in xenografts. Peptide-camptothecin (CPT) conjugate is claimed to be selectively cytotoxic to the cancer cells using a xenograft and PDX model. I have the following concerns that need to be addressed before it can be published.

1. I am surprised by the lack of specific sensitivity of P-LPK-CPT conjugate in Figure 4A-B? There doesn't seem to be added selectivity from P-LPK-CPT conjugate compared to the CPT control. While there seems to be higher sensitivity in HCT116 cells in Supplementary Figure 6. Is this due to the PDX model not expressing high levels of SLC1A5? Can this be repeated in another PDX derived line?

Another option could be, testing the highest SLC1A5 expressing HT29 line and shRNA knockdown of SLC1A5 (supplementary 10G). This will strengthen the sensitivity and selectivity of the peptide-conjugate and greatly strengthen the manuscript, as the data provided in 5E, F although striking are preliminary.

2. Authors claim the uptake and membrane localization of P-LPK shown in Figure 1A and 1E are significantly higher in P-LPK treated cells. But the images shown are not clear, they need to be improved, and include higher magnification images of all lines used. Same for Supplementary Figure 2A, 5A.

3. Could the authors comment on the cross-reactivity of the peptide to SLC1 family members?

4. Cell viability measurements throughout the manuscript are shown as OD450 values, can the authors provide them relative to the DMSO/control (showing all data points on the plots), this will make it comparable across cell types used. See 3E and 3F.

5. microPET in Figure 2A-C showing ⁶⁸Ga-P-LPK does not show quantification for all cell lines used. Please provide quantification for all.

Minor

1. Supplementary 7b is showing small intestine but has been mislabelled as colon. Please add a panel with HE of colon of the same treatment groups.

2. Suggestion to replace Figure 4E with IHC of Cleaved. Caspase 3 or cl. Parp. These are easier to quantify and present than IF images shown currently.

Dear Editors and Reviewers:

It is highly appreciated for your constructive comments concerning our article. We would like to express our gratitude to all the reviewers for their valuable and helpful comments. We have studied comments carefully and tried best to incorporate all of their suggestions into the revised manuscript. In this revised version, changes to our manuscript are marked in red. Herein, we have addressed all the comments (in black) with a point-by-point response (in blue). We hope, therefore, that the revised manuscript is acceptable for publication in *Communication Biology*. As detailed below, we now address each point individually.

Reviewers' comments:

Reviewer #1 (Remarks to the Author):

The manuscript deals with an interesting topic that is the role of asct2 in CRC and with the potential therapeutic effects of a novel peptide. this is, of course, very interesting for both basic and applied research. However, some concerns arose which are listed below:

1) in general some sentences are not very clearly written such as pg 2 lines 50-51.

Response:

We thank the reviewer for this helpful suggestion to improve our manuscript. We have revised some sentences (including pg 2 lines 50-51) and marked them in red. We are sure that our paper will be more easily read according to your suggestion.

“The phage display technique is a powerful tool to identify high-affinity peptides for selected targets that present interesting in researches and clinics.” in pg 2 lines 50-51 have been revised to “The phage display technique is a powerful tool to identify novel targeting peptides that is interested in researches and clinics.”

2) the hypothesis on the role of asct2 is not clear. what do the authors suggest? is asct2 responsible for binding the peptide or for transporting it? is the transport mediated by the classical elevator mechanism of asct2 or do the authors hypothesize an endocytotic mechanism based on ASCT2? Consider that ASCt2 is considered a viral receptor not only a transporter, therefore the endocytotic mechanism needs to be taken into consideration. Indeed, the transport of a long peptide instead of a single amino acid is a bit difficult to think considering the mechanism of membrane transporters.

Response:

Thank you for your insightful and professional suggestion. We agree with the reviewer that a long peptide is a bit difficult to transport considering the mechanism of membrane transporters. Macromolecule entry into cells is mainly through different endocytic mechanisms, such as clathrin-mediated endocytosis, macropinocytosis. We hypothesized that the peptide P-LPK binds to SLC1A5, which is up regulated in colorectal cancer cells, and then enters the cell through endocytosis.

To test this hypothesis, we performed endocytosis inhibition experiments. HCT116 cells were

treated with various endocytotic inhibitors, including chlorpromazine (CPZ, inhibiting clathrin-mediated endocytosis), methyl-beta-cyclodextrin (M β CD, inhibiting lipid raft-mediated endocytosis) and amiloride (EIPA, inhibiting macropinocytosis). The fluorescence intensity was measured in a multi-well plate reader (BioTek synergy4). The results showed that chlorpromazine exhibited the highest inhibitory effect. Chlorpromazine significantly decreased the cellular internalization of FITC-P-LPK, suggesting that uptake of P-LPK was predominantly via a clathrin-mediated mechanism. The figure has been included in the Figure 6 C-D (shown below). Correspondingly, we added the text in the main text.

It is important to recognize that most endocytosis inhibitors are working on multiple pathways, which makes it difficult to draw definitive conclusions about the endocytic pathways. In future studies, we plan to use more specific approaches, such as knockout of key components in the internalization pathways to investigate endocytotic mechanism.

3) the expression levels of ASCT2 in the tested cells lines should be demonstrated and shown by western blot analysis.

Response:

The comments are important. The expression levels of ASCT2 in the tested cells lines have been demonstrated and shown by western blot analysis. We have added new data in Figure 6A.

4) a proof of peptide specificity, followed by docking, would be the use of a mutant peptide in those residues considered responsible for interaction with ASCT2.

Response:

Thank you for your rigorous comment. P-LPK has been mutated (Sequence APATVSSDMSLA) and the interaction with ASCT2 is detected by confocal microscopy (Figure 5I). The fluorescence intensity of FITC labeled mutant peptide (FITC-P-MUT) bound to HCT116 cells was significantly reduced compared to FITC-P-LPK.

5) docking analysis should be redone because the authors used the 5lm4 structure that is not asct2 but eaat1. the cryoem structures of asct2 are available in different conformations, consider as an example pdb 6gct or others from the same authors.

Response:

Thanks reviewer for careful examination. We have corrected the 5lm4 to 6gct and redone the docking in this revised version used asct2 (shown in Figure 5C-F). Correspondingly, we modify the text below in the main text.

In the main text:

The 3D structure models of the P-LPK peptide and SLC1A5 are shown in Figure 5C. Two main kinds of interactions were observed: The side chain of the M267, S111, A423 and S354 formed hydrogen bonds with the L1, K3 and N12 in the P-LPK peptide. The residues L1, K3 and N12 in the P-LPK peptide inserted into the hydrophobic pocket formed by the L112, G115, S118, L119, I381, T384, V385, V426, G427, and A429 residues (Figure5 D-F).

6) another important experiment that could be done is the competition experiment with asct2 substrates to help distinguish between actual transport and another mechanism such as endocytosis.

Response:

We accept your thoughtful and professional comments. We have performed the competition experiment with asct2 substrates glutamine. FITC-P-LPK was incubated with increasing concentrations (0-32mM) of glutamine in HCT116 cells at 37°C for 2h. Then these cells were washed with ice-cold PBS three times gently, suspended in 200µl PBS and transferred to a 96-well costar plate. The fluorescence intensity was measured in a multi-well plate reader. The results showed that the FITC fluorescence intensity did not change with increasing glutamine concentration, suggesting that glutamine does not affect the uptake of P-LPK. Results are shown below (Figure 6E).

Reviewer #2 (Remarks to the Author):

This manuscript studies a peptide, P-LPK, which was identified from the phage-display library toward colorectal cancer cells (CRC). This study validates that P-LPK peptide binds to CRC cells in cell culture as well as in mice models. The authors further conjugate the peptide with camptothecin (CPT), a topoisomerase inhibitor as a chemotherapy agent. The P-LPK-CPT conjugate has cytotoxicity toward CRC cancer cells both in vitro and in vivo. The target of P-LPK was also investigated, and SLC1A5 is a potential target based on Co-IP and protein structure modeling. Overall, this is an interesting manuscript with the potential for future clinical application. However, the novelty is slightly compromised by the previous report showing another peptide, CBP-DWS, toward CRC from the same research group (Hou et al, 2018). Some of the experimental details or explanations are missing. The suggestions and questions are listed below.

1. The result that CPT alone is toxic to mice while P-LPK-CPT has an anti-tumor effect is interesting. The authors speculate that CPT is hydrolyzed from P-LPK-CPT by esterase in CRC. However, esterase activity is abundant in serum, for example, albumins have been shown to exhibit esterase activity. Can the authors provide explanations for why the P-LPK-CPT will not be hydrolyzed in blood and caused toxicity from CPT before the peptide reaches the tumor

microenvironment?

Response:

Thank you for your comments. Ester bonds are not only hydrolyzed by esterase but also sensitive to acidic environments. The rapid growth and metabolism of tumor cells result in lower pH, higher contents of many enzymes than blood. Besides, the intracellular compartments like endosomes and lysosomes of cancer cells are rich in esterase. Hence, P-LPK-CPT was hydrolyzed not only by the esterase that is highly expressed in CRC cells but also favored by the acidic tumor microenvironment.

However, as you mentioned, albumins exhibit esterase-like activities. Indeed, ester bonds are prone to hydrolysis in serum, so their extracellular stability needs to be carefully controlled. Therefore, we carry out a new experiment to assess the plasma stability (shown below). The results showed that P-LPK-CPT displayed high plasma stability, which was attributed to the high linker stability in blood.

Method:

The stability behavior of P-LPK-CPT was investigated under a simulated physiological condition at 37 °C. A total of 3 mL of P-LPK-CPT was transferred into a dialysis bag (MWCO = 3000 g/mol). The dialysis bag was put in the flask, immersed by 60 ml of pH 7.4 phosphate buffer solutions containing plasma and stirred slightly at 37° C in the dark. Then, 2 mL of the internal PBS buffer was replaced with 2 mL of fresh PBS immediately at predetermined time intervals, keeping the sinking condition. Here, the fluorescence intensity (Ex: 340 nm, Em: 430 nm) of the internal buffer was used to analyze the amount of CPT that was released from the conjugate.

2. Please soften the claim on title “targeting SLC1A5” and the last sentence of abstract “via targeting SLC1A5”. Since the current data only support that P-LPK may bind to SLC1A5. If the author would like to keep the claim, please provide data that genetically or chemically inhibiting SLC1A5 did abolish the anti-tumor effect of P-LPK-CPT, while overexpression of SLC1A5 sensitizes the CRC cells to P-LPK-CPT.

Response:

Thank you very much for your insightful suggestion. We have softened the claim on title “targeting SLC1A5” and the last sentence of abstract “via targeting SLC1A5”. The title was changed to

“P-LPK-CPT, a peptide-camptothecin conjugate, is highly effective for colorectal cancer”. The abstract was accordingly modified and marked in red. And we will carry out new experiments to support the claim “targeting SLC1A5” in future studies.

3. Why the authors did not choose to conjugate P-LPK with the major CRC chemotherapy choices, such as 5-FU, oxaliplatin, and irinotecan?

Response:

Thank you for your valuable comment. We conjugated P-LPK with 5-FU, but were unsuccessful and we will try to improve the method in future. Conjugating P-LPK to oxaliplatin by different methods is ongoing in our Lab and the results also show enhanced anti-tumor effects, data are being collated and preparing for submission.

4. What does CBP- or P- stand for in the name of the two peptides, CBP-DWS and P-LPK?

Response:

We apologize for the confusion. CBP is the abbreviation of “Colorectal cancer Binding Peptide”, P is the abbreviation of “Peptide”. The use of “P” instead of “CBP” was to make writing more concise, so that it is convenient for us to show in the figure (Such as ⁶⁸Ga-P-LPK, P-LPK-CPT).

5. It is a puzzle why CBP-DWS-CPT doesn't have a cytotoxicity effect (Figure S2). Both CBP-DWS and P-LPK bind to CRC cells, but only P-LPK-CPT has a cytotoxicity effect. Can the author discuss the reasons that cause the discrepancy?

Response:

Thanks for your critical comments. In general, peptide-drug conjugate (PDCs) are composed of peptides, drugs and linkers. Conjugating peptide to drugs requires chemical groups on both of the peptide and the drug that are compatible for chemical conjugation. Therefore, a linker has to be designed properly to avoid damaging the targeting ability of peptide or/and the drug's anti-tumor activity. Here, CBP-DWS-CPT doesn't have a cytotoxicity effect, indicating that this linker is not successful for CBP-DWS and CPT. We will try different methods to link CPT and CBP-DWS in further study.

6. Are Figure 1A and Figure S3A live images or after fixation? From the method section, it seems they are live images while the DAPI is listed. As we know, DAPI is used for fixed samples. What is the concentration of DAPI used?

Response:

Thank you for your detailed review and we totally agree with you. DAPI is best used for fixed samples. We apologize for our careless. The description of the procedure is unclear. The cells were fixed with 4 % paraformaldehyde for 30 min at room temperature, and then DAPI (1µg/ml) was used for nucleic acid stain. Modifications have been made accordingly in the method section.

7. The fluorescence signals of SW480 P-CON vs P-LPK in Figure S3A are not very different, while the quantification data (Figure S3B) shows a significant difference. Please show representative images or please indicate how the quantification was processed. Also, please indicate if the microscopy setting and image process are identical between experimental and control groups.

Response:

We appreciate the reviewer's comment on the image. It may be due to the low resolution that the picture was not displayed clearly. We have showed a new image of SW480 in Figure S3A. The microscopy setting and image process are identical between experimental and control groups (using the same laser power and exposure time).

8. Please provide the catalog number of the Thermo “blue membrane probe”.

Response:

The catalog number of the Thermo “blue membrane probe” is “Thermo Fisher Scientific, Cat. No. W11263”.

9. I am confused that the authors first claimed the peptide sequence of P-LPK is unique (supplementary table 2), but later they found P-LPK has a fairly homologous sequence with Trm1 (75%).

Response:

We appreciate your comment and apologize for confusion. We thought the peptide sequence of P-LPK is unique, because no known protein showed 100% homologous sequences with the peptide P-LPK (twelve amino acids) was found by BLAST (Supplementary table 2). Eight of twelve amino acids (8/12) in the P-LPK are identical to Trm1 (shown below). Despite about 67% (8/12, we are sorry it's about 67%, not 75%) homologous sequences, they are not exactly the same.

Identical sequence is highlighted

Trm1 complete sequence VS P-LPK peptide (Sequence LPKTVSSDMSLN)

MEIVQEGIAKIIVPEI**PKTVSSDM**PVFYNPRMRVNRDLAVLGLEYLCKKLGRPVKVADPLS
ASGIRAIRFLLTSCVEKAYANDISSKAIEIMKENFKLNNIPEDRYEIHGMEANFFLRKEWG
FGFDYVDLDPFGTPVPFIESVALSMKRGGILSLTATDTAPLSGTYPKTCMRRYMARPLRNE
FKHEVGIRILIKKVIELAAQYDIAMIPIFAYSHLHYFKLFFVKERGVEKVDKLIQFGYIQYC
FNCMNREVVTDLKYFKEKCPHCGSKFHIGGPLWIGKLWDEEFTNFLYEEAQKREEIEKET
KRILKLIKEESQLQTVGFYVLSKLAEKVKLPAQPPIRIAVKFFNGVRTHFVGDGFRTNLSFE
EVMKKMEELKEKQKEFLEKKKQG

10. Please show the data that P-LPK insets into the hydrophobic pocket (line 216, not shown).

Response:

Thank you for your kind suggestion. We have showed the data that P-LPK insets into the hydrophobic pocket in Figure 5D. The binding pocket of SLC1A5 was represented by the electrostatic potential, where P-LPK was shown in pale blue stick.

11. Please show the data to support the sentence “ however, ANXA3 displayed much lower affinity...and the K74, K98, K102, and Y137 residues”. (line 218-220).

Response:

Thank you for your comment. The possible minimum-energy conformations of the peptide P-LPK binding sites of ANXA3 is higher than that of SLC1A5 (ΔG -5.9317 kcal/mol vs -6.7770 kcal/mol), suggesting that ANXA3 displayed much lower affinity with P-LPK.

We further examined the affinity of ANXA3 to P-LPK by SPR (shown below). The results showed that the K_D is $676 \times 10^{-6} M$, demonstrating that ANXA3 has a lower affinity for P-LPK than SLC1A5 (K_D , $1.19 \times 10^{-6} M$).

12. Which GEO database (accession number) is used for figure S10F?

Response:

The GEO database (accession number) is GSE41258. We have added accession number in revised manuscript.

13. Please provide reference or data to support the statement “SLC1A5 is down-regulated in malignant glioblastoma”.

Response:

Thank you for your detailed review. Our expression here is imprecise. In fact, some literatures reported that SLC1A5 (ASCT2, encoded by SLC1A5 gene) is up-regulated in malignant glioblastoma^{1,2}. In this study, we detected SLC1A5 in malignant glioblastoma cell U87, and found the expression level of SLC1A5 in U87 cells was significantly lower than HCT116 cells (shown below). This result is consistent with the microPET imaging. We had revised the statement “SLC1A5 is down-regulated in malignant glioblastoma” and marked it red in discussion section.

1. Syafruddin S, Nazarie W, Moidu N, et al. Integration of RNA-Seq and proteomics data identifies glioblastoma multiformer surface signature. BMC cancer 2021;21:850.
2. Sidoryk M, Matyja E, Dybel A, et al. Increased expression of a glutamine transporter SNAT3 is a marker of malignant gliomas. Neuroreport 2004;15:575-8.

14. Typo errors: (1) line 354: DBCO- instead of DCBO- (2) line 366 and 367: ddH2O instead of ddH2O2.

Response:

We sincerely apologize that we made such a blunder. We have corrected “DCBO-” to “DBCO-” and “ddH₂O₂” to “ddH₂O”.

15. Figure 2 legend mentions that kidneys uptake ^{68}Ga -peptide. Out of curiosity, why do the kidneys uptake a prominent amount of ^{68}Ga -peptide? Also, it seems that the ^{68}Ga -peptide accumulates at multiple tissues, not just the kidney (heart? Liver? Bladder?)

Response:

We thank the reviewer for pointing this out. The prominent uptake of ^{68}Ga -peptide was observed at the bladder (not kidney) at early time (30min), suggesting that this tracer may be excreted through the urinary route. The reviewer is right; ^{68}Ga -peptide accumulates at other tissues with time extending gradually (60min, 120min). We have made changes accordingly (marked in red in Figure 2 legend).

16. Considering that CPT is a chemo drug and it induces toxicity in mice, wouldn't it be surprising that CPT alone and control-peptide-CPT don't have cytotoxicity effects (Figure 3D-3E)? Could the authors discuss more this finding?

Response:

Thank you for your careful review. We tested the anti-tumor effect of P-LPK-CPT at different concentrations on CRC cells and found that P-LPK-CPT exhibited significant antitumor activity at very low concentrations (10nM). But at this low concentration (with equivalent CPT dose), CPT alone and control-peptide-CPT exhibits no obvious toxicity to CRC cells (Figure 3D-3E). Both CPT alone and control-peptide-CPT had obvious cytotoxicity effects at 100nM (shown below).

17. It is not fair to state that PI is a nuclear stain (Figure S5A figure legend) because PI also stains mitochondria DNA.

Response:

Thank you very much for your remind. We have corrected “nuclear stain” to “nucleic acid stain” in Figure S5A figure legend.

18. The figure S10G is quite important and it may be moved to the main figure. Since Colo320HSR has the highest signal of P-LPK in figure 1A, does the cell line also have a high expression of SLC1A5?

Response:

We appreciate your helpful suggestion. We have moved the figure S10G to the main figure as

Figure 6A. We detected SLC1A5 in Colo320HSR, the result showed that SLC1A5 was also highly expressed in Colo320HSR (Figure 6A).

Reviewer #3 (Remarks to the Author):

In this manuscript, the authors report the identification of a peptide-drug conjugate that specifically targets amino acid transporter SLC1A5 in colorectal cancer cells. Using a phage-display system the authors identify and validate the *in vitro* and *in vivo* safety profile and applicability of the novel peptide. They go on to show that the peptide specifically binds to cancer cells and accumulated in xenografts. Peptide-camptothecin (CPT) conjugate is claimed to be selectively cytotoxic to the cancer cells using a xenograft and PDX model. I have the following concerns that need to be addressed before it can be published.

1. I am surprised by the lack of specific sensitivity of P-LPK-CPT conjugate in Figure 4A-B? There doesn't seem to be added selectivity from P-LPK-CPT conjugate compared to the CPT control. While there seems to be higher sensitivity in HCT116 cells in Supplementary Figure 6. Is this due to the PDX model not expressing high levels of SLC1A5? Can this be repeated in another PDX derived line? Another option could be, testing the highest SLC1A5 expressing HT29 line and shRNA knockdown of SLC1A5 (supplementary 10G). This will strengthen the sensitivity and selectivity of the peptide-conjugate and greatly strengthen the manuscript, as the data provided in 5E, F although striking are preliminary.

Response:

Thank you for your thoughtful and professional comment. We appreciate for your kind suggestion. As suggested, we knockdown SLC1A5 in HT29 and the FITC-P-LPK showed a significant reduction of fluorescence. This result further validated the role of SLC1A5 in the binding of P-LPK to CRC cells. We have added new data in Figure 6B (shown below).

Then, we detected SLC1A5 expression levels in three CRC tumor tissues and found the up regulation level of SLC1A5 varies widely in different tissues (T1 was used in PDX model in this paper). Thus, P-LPK-CPT deserves further development as a promising anticancer therapeutic for CRC treatment, especially SLC1A5-high expression CRC type. We will investigate P-LPK-CPT conjugate in different PDX derived line in further study.

2. Authors claim the uptake and membrane localization of P-LPK shown in Figure 1A and 1E are significantly higher in P-LPK treated cells. But the images shown are not clear, they need to be improved, and include higher magnification images of all lines used. Same for Supplementary Figure 2A, 5A.

Response:

We thank the reviewer for pointing this out, which may be due to file compression during submission. We have adjusted all images in the manuscript to the best resolution as we can, and will upload individual uncompressed figures with suitable format to keep the high quality of images. Higher magnification images have also been used in Figure 1A , 1E and Supplementary Figure 3A, 5A (shown below).

Figure 1A

Figure 1E

Supplementary Figure3A

Supplementary Figure5A

3. Could the authors comment on the cross-reactivity of the peptide to SLC1 family members?

Response:

Thank you for your insightful suggestion to improve our manuscript. We have added some of them in Discussion section in the revised manuscript.

The solute carrier family 1 (SLC1) consists of five high-affinity glutamate transporters EAAT1-EAAT5 and two neutral amino acid transporters ASCT1 and ASCT2 (ASCT2, encoded by gene SLC1A5). Although SLC1 family members have similar predicted structures, they exhibit distinct functional properties. In addition to its role as a transporter, SLC1A5 is also a receptor for many retroviruses, including simian retrovirus 4, feline endogenous virus, etal. Here, we found SLC1A5 might be the receptor of P-LPK. Theoretically, P-LPK may have cross binding ability to other SLC1 families, but further experimental verification is required.

4. Cell viability measurements throughout the manuscript are shown as OD450 values, can the authors provide them relative to the DMSO/control (showing all data points on the plots), this will make it comparable across cell types used. See 3E and 3F.

Response:

It is a very valuable comment. We have provided values relative to the DMSO/control and showed all data points on the plots in Figure 3E and 3F.

5. microPET in Figure 2A-C showing ^{68}Ga -P-LPK does not show quantification for all cell lines used. Please provide quantification for all.

Response:

Thank you for your nice suggestion. We have showed the quantification values in the lower right corner of mice. Besides, in the caption of Figure 2A, we have added one sentence “The quantification values of tumor were showed in the lower right corner of mice”.

Minor

1. Supplementary 7b is showing small intestine but has been mislabelled as colon. Please add a panel with HE of colon of the same treatment groups.

Response:

Thank you for your correction. We have changed the colon to small intestine (ileum) in Supplementary 7B. We are very sorry that we did not collect colon for HE staining in this study.

2. Suggestion to replace Figure 4E with IHC of Cleaved. Caspase 3 or cl. Parp. These are easier to quantify and present than IF images shown currently?

Response:

We gratefully appreciate for your valuable suggestion. However, it has been a long time since the experiment, and we found the tissues structure has been destroyed. Therefore, IHC are not performed successfully. We will use IHC in our future research.

REVIEWERS' COMMENTS:

Reviewer #1 (Remarks to the Author):

the authors addressed my major concerns.
therefore, from this study it can be concluded that ASCT2 helps the entry of the peptide that, however, does not cross membrane as a "substrate" of ASCT2 considering that glutamine does not compete with the peptide for the binding to ASCT2.
this has to be made clear for readers

Reviewer #2 (Remarks to the Author):

I would like to thank the authors for the revision. The authors have addressed all my comments and suggestions. I have no further comments.

Reviewer #3 (Remarks to the Author):

The authors have addressed the comments to my satisfaction. I would suggest the add the SLC1A5 cross-reactivity from my review point 3 in the discussion. Recommend for publication.

Dear Editors and Reviewers,

Thank you very much for your hard work and further consideration on our manuscript. We also thank all the reviewers for their positive and encouraging comments. We have revised the manuscript in accordance with the reviewers' comments (marked in red) and edited this manuscript according to the editorial requests. We strive to impress the readers more comprehensive and important information. We hope that all these changes fulfill the requirements to make the manuscript acceptable for publication in Communication Biology.

REVIEWERS' COMMENTS:

Reviewer #1 (Remarks to the Author):

The authors addressed my major concerns.

therefore, from this study it can be concluded that ASCT2 helps the entry of the peptide that, however, does not cross membrane as a "substrate" of ASCT2 considering that glutamine does not compete with the peptide for the binding to ASCT2. this has to be made clear for readers.

Response:

Thank you for your professional comments that help to deepen the understanding of this study.

We have added this conclusion in discussion section to make the study clearer for readers.

Reviewer #2 (Remarks to the Author):

I would like to thank the authors for the revision. The authors have addressed all my comments and suggestions. I have no further comments.

Response:

Thank for your hard work and for positive comments.

Reviewer #3 (Remarks to the Author):

The authors have addressed the comments to my satisfaction. I would suggest the add the SLC1A5 cross-reactivity from my review point 3 in the discussion. Recommend for publicaiton.

Response:

We thank the reviewer for this suggestion to improve our manuscript. We have added the SLC1A5 cross-reactivity from review point 3 in the discussion.